# STUMP Swiftly Followed by Large Adenomyoma in a Young Nulliparous Patient

**DOI:** 10.3390/diagnostics15233018

**Published:** 2025-11-27

**Authors:** Georgiana Nemeti, Gheorghe Cruciat, Iulian Gabriel Goidescu, Chereches Roberta, Vasile Marian Ticala, Mihai Surcel, Cerasela Mihaela Goidescu, Adelina Staicu, Dan Boitor-Borza, Bogdan Fetica, Ioana Cristina Rotar, Daniel Muresan

**Affiliations:** 1Obstetrics and Gynaecology I, Mother and Child Department, University of Medicine and Pharmacy “Iuliu Hatieganu”, 400006 Cluj-Napoca, Romania; georgiana.nemeti@elearn.umfcluj.ro (G.N.); goidescu.iulian@elearn.umfcluj.ro (I.G.G.); chereches.roberta@gmail.com (C.R.); mihai.surcel@elearn.umfcluj.ro (M.S.); adelina.staicu@elearn.umfcluj.ro (A.S.); dan.boitor@elearn.umfcluj.ro (D.B.-B.); cristina.rotar@umfcluj.ro (I.C.R.); muresandaniel01@elearn.umfcluj.ro (D.M.); 2Radiology Department, Cardiomed Health Center, 400015 Cluj-Napoca, Romania; marianticala@yahoo.com; 3Department of Internal Medicine, Medical Clinic I—Internal Medicine, Cardiology and Gastroenterology, University of Medicine and Pharmacy “Iuliu Hatieganu”, 400006 Cluj-Napoca, Romania; sava.cerasela@elearn.umfcluj.ro; 4Department of Pathology, “Prof. Dr. Ion Chiricuta” Institute of Oncology, 400015 Cluj-Napoca, Romania; fetica.bogdan@elearn.umfcluj.ro; 5Molecular Pathology Laboratory Pathos, 400394 Cluj-Napoca, Romania

**Keywords:** AUB, nullipara, “in status nascendi”, STUMP, large adenomyoma, hysterectomy

## Abstract

The potential concurrence of uterine leiomyoma and adenomyosis has been mentioned in several studies to date, but as co-existing entities, not as a sequence of pathologic events. This is the case of a young 31-year-old nulliparous patient presenting with unspecific pain and abnormal uterine bleeding (AUB) pattern symptoms, which was clinically diagnosed with FIGO 0 fibroid “in status nascendi”. Following removal, the tumor turned out to be a STUMP at the histopathologic workup. After 9 months postoperatively and two unremarkable follow-ups, the patient presented again for pelvic pain and AUB, when ultrasound revealed a heterogeneous endo-uterine tumor of 5 cm, rich in large vessels, with rapid growth at serial ultrasound. MRI established the diagnosis of adenomyosis. In the context of a prior STUMP, nulliparity and rapidly enlarging uterine mass, despite conservative management counseling in a multidisciplinary team, the patient preference was towards radical surgery to prevent any future reproductive organ-related distress, and she opted for total hysterectomy with bilateral adnexectomy. The co-existence of fibroid and adenomyosis has been signaled by several authors, but this is the first report of such a sequence of events (STUMP to large adenomyoma) with swift development, to the best of our knowledge.

**Figure 1 diagnostics-15-03018-f001:**
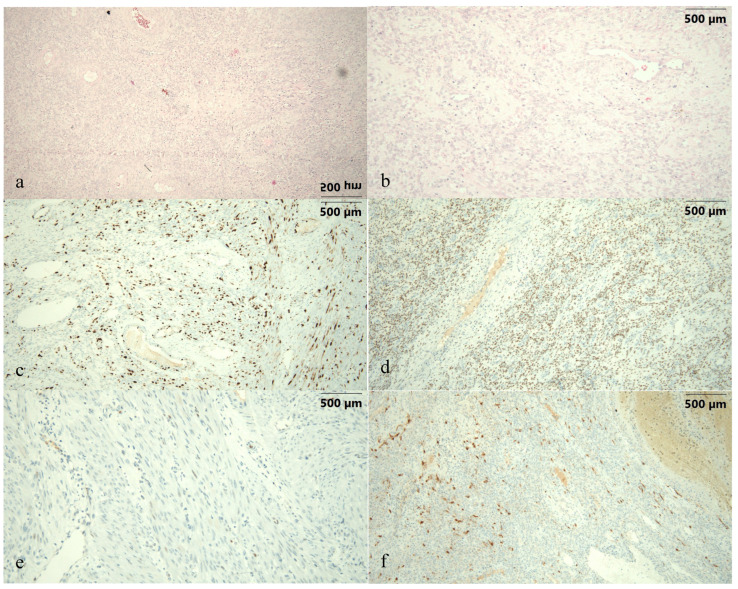
This is the case of a 31-year-old G0 P0 female patient with no relevant personal and family history and negative screening for cervical cancer initially attending gynecologic care for strong dysmenorrhea, abnormal uterine bleeding (AUB) pattern, irregular menstrual cycles, foul-smelling discharge, lower abdominal pain, weakness and fatigue. The clinical gynecological examination identified a vaginal leiomyoma in “status nascendi”, FIGO 0 (International Federation of Gynecology and Obstetrics) [1], of about 5 cm in size and confirmed the malodorous discharge. The transvaginal ultrasound examination revealed a normally appearing uterus and adnexa and the intravaginal tumor with ultrasound characteristics of leiomyoma (images unavailable). Laboratory workup was unremarkable. Following counseling of the patient and obtaining informed consent, she was scheduled for myomectomy by torsion in a vaginal approach, followed by biopsy curettage of the endometrial cavity. Pathology resulted in a smooth muscle tumor of uncertain malignant potential (STUMP) with positive immunostaining characteristics, as shown in Figure 1. Images present (**a**,**b**) hematoxylin–eosin staining slides showing dense cellular proliferation, cytoarchitectural atypia and atypical mitoses; images (**c**–**f**) present immunohistochemistry staining slides depicting an increased mitotic rate and proliferation index (**c**) Ki 67–30%, (**d**) positive estrogen receptor (ER) of 50% with weak intensity, (**e**) negative p53 and (**f**) focally positive p16. At the 2-month postoperative follow-up, all patient complaints had been resolved and both the clinical and ultrasound evaluation were in the normal range, as presented in the case timeline (Figure 2). Given the pathology result, the patient was recommended to perform contrast-enhanced pelvic MRI (magnetic resonance imaging) and oncological evaluation. Given the MRI scan was normal, the oncologist recommended biannual evaluation. Another normal clinical evaluation and examination were confirmed at the 6-month follow-up.

**Figure 2 diagnostics-15-03018-f002:**
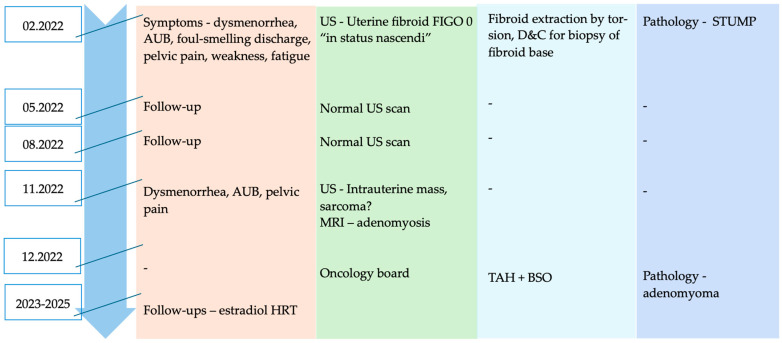
Case timeline highlighting patient presentation time-points and actions taken (AUB—abnormal uterine bleeding; US—ultrasound; D&C—dilation and curettage; STUMP—smooth muscle tumor of uncertain malignant potential; MRI—magnetic resonance imaging; TAH + BSO—total abdominal hysterectomy with bilateral salpingo-oophorectomy; HRT—hormonal replacement therapy).

**Figure 3 diagnostics-15-03018-f003:**
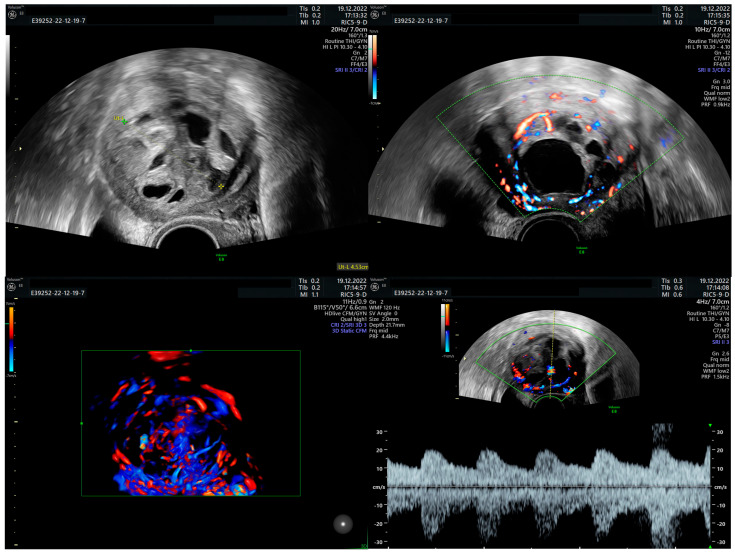
After 9 months postoperatively the patient registered for another gynecological evaluation, presenting again with AUB and strong dysmenorrhea. The clinical examination was normal, but the transvaginal ultrasound revealed the presence of a roughly 3.5 cm heterogeneous intrauterine tumoral mass, with enlarged vessels and rich Doppler signal with low velocity flow. Both ovaries had a regular ultrasound appearance. During the interval of approximately two weeks when the patient sought several medical opinions, tumor size increased to 5 cm when she arrived in our service.

**Figure 4 diagnostics-15-03018-f004:**
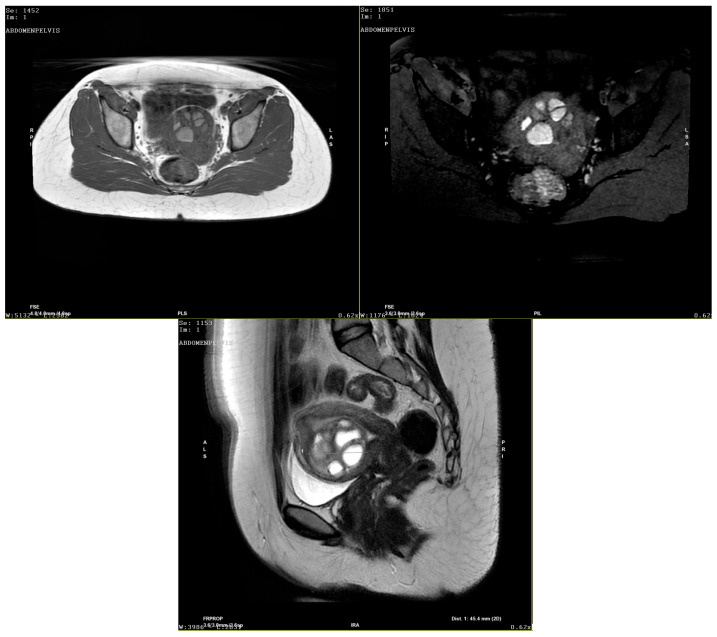
The contrast-enhanced pelvic MRI scan describes the 42/42/49 mm uterine mass as a macronodular intracavitary tumor with a round contour, with a retro-tumoral 4.5 mm mural width to the uterine serosa. The mass showed a mixed solid fibroglandular and necrotic liquid component in T hypersignal, hemorrhagic in T1 hypersignal, which persisted in T1 FS sequence, with socket peripheral contrast. The radiologist diagnosed the tumor as intracavitary adenomyoma. Given the previous surgery with a STUMP pathology result and the rapid growth of the tumor, the suspicion of uterine sarcoma was also raised, and the case was sent to an oncology board. Conservative surgery—tumorectomy—was recommended, considering the patient had no pregnancies, followed by a regular 3-month follow-up schedule. However, upon counseling, the patient argued that she had no desire to bear children and rejected the possibility of taking any risks or other surgeries and opted for a more radical approach—total hysterectomy with bilateral adnexectomy (gross pathology images not available). The pathology examination confirmed the intrauterine mass to be a giant adenomyotic structure (images not available). The patient received postoperative hormonal substitution therapy and is currently under multidisciplinary surveillance given the surgical menopause. This case is special due to several reasons: STUMP pathology diagnosed in a 30-year-old patient, that is, younger than the largely reported age interval for this disease; STUMP pathology depicted in a “status nascendi” FIGO 0 fibroid; and rapid post-excisional development of subsequent giant intrauterine adenomyoma. STUMPs represent a category of mesenchymal tumors of the uterus, also known as “borderline” uterine masses since they can behave both as benign fibroids and also act as malignant uterine leiomyosarcomas having metastases, recurrence and generally poor prognosis [2,3]. The classification of uterine mesenchymal tumors remains difficult due to significant overlap among entities such as STUMPs and various forms of atypical leiomyoma. STUMPs represent a diagnostic, counseling and management challenge. Once the suspicion is raised, given their rare occurrence and lack of standardization and management guidelines, it is difficult to counsel patients regarding prognosis, especially in those of reproductive age who desire pregnancy [4]. Most STUMPs are diagnosed retrospectively in surgery specimens following myomectomy of hysterectomy for uterine fibroids. The age of presentation ranges between 40 and 50 years old, similarly to uterine fibroids/sarcomas [5]. Presenting symptoms range from silent, asymptomatic tumors to AUB patterns with/without secondary anemia or bulk symptoms such as any pelvic mass. Tumor markers, even CA125 and LDH (lactate dehydrogenase), have not proven their value in diagnosing STUMPs or uterine malignancy [6]. In imaging studies, neither ultrasound nor MRI are foolproof in establishing the diagnosis. Ultrasound may describe an atypically looking uterine mass, more heterogeneous than a fibroid, with more intense vascularization and hypoechoic uterine wall nodules. However, studies report inconsistent correlations between these parameters and pathologic samples [7,8,9]. MRI provides better soft tissue resolution; however, even with the use of diffusion-weighted imaging, apparent diffusion coefficient values and contrast-enhanced MRI imaging [10], a definitive differential diagnosis cannot be achieved [11]. This is also due to the similar MRI characteristics of cellular type of leiomyomas and different types of degenerative transformation types/necrosis which may be encountered in leiomyomas [5,6]. The role of PET-CT remains uncertain; although 18F-FDG uptake can help distinguish leiomyosarcomas from fibroids, it is generally unreliable for differentiating leiomyosarcomas from STUMPs, both having a “hollow ball” sign due to a zone of typical coagulative necrosis [12]. The reported recurrence rate of STUMPs is 7 to 36.4%, usually occurring as either another STUMP or benign leiomyoma, but occurring approximately 5 years after primary surgery [13,14]. This would come into contradiction with the rapid regrowth from our case; however, given the first histology, one would be prone to suspect disease flare-up. Malignant transformation of uterine adenomyosis is exceedingly rare, with fewer than 50 cases reported in the medical literature [15]. Most involve endometrioid or clear cell carcinoma, while serous carcinoma is exceptionally uncommon—only five reports describe five serous carcinomas and six serous endometrial intraepithelial carcinomas (EICs) to date [15,16,17,18,19,20]. Furthermore, Abushahin et al. identified a p53 mutation in one of five serous EICs arising in adenomyosis and suggested that EmGD and serous EIC in adenomyosis may be underdiagnosed due to their minute foci [17]. Supporting this view, Lu et al. noted that serous carcinomas in adenomyosis may be under-recognized, especially when concurrent carcinoma is present in the endometrium or extrauterine sites [15]. The co-occurrence of adenomyosis and leiomyoma has been reported before by several authors [21,22,23]. We can speculate that certain patients develop more types of uterine pathologies, perhaps through yet undiscovered pathological underpinnings. This could be one potential explanation for the development of two types of uterine pathologies in such a rapid sequence in our patient. Nava et al. report the finding of a focus of adenomyosis inside an atypical, highly cellular leiomyoma [24]. Adenomyosis, represented by the invagination of endometrial stroma/glands into the myometrium of the uterus, may be diagnosed as focal or diffuse disease. However, such large, heterogeneous tumors as those described in our patient are uncommon, even disregarding the potential association with the previous STUMP [25]. Atypical, large adenomyomas previously reported have taken the form of cystic adenomyomas [26,27,28,29,30,31]. Also, interestingly, authors report such adenomyomas to be more frequent in younger patients [30]. The mass from our case was circumscribed—which is not a feature suggestive of adenomyoma—heterogeneous, with numerous, large, low-flow vessels. Considering the background of the young, nulliparous patient with previous STUMP surgery, rapid tumor growth and atypical imaging findings, this was a difficult case to diagnose, counsel and manage. The co-existence of fibroid and adenomyosis has been signaled by several authors, but this is the first report of such a sequence of events (STUMP to large adenomyoma) with fast development (within a few months and weekly growth), to the best of our knowledge. The concurrent or subsequent development of uterine pathologies is not a rare event (fibroids, polyps, adenomyosis). However, the transition from one type of lesion to another is rare, and the only transition better accounted for is fibroid/STUMP to sarcoma. The swift transition from fibroid/STUMP to adenomyosis has not yet been reported, without assuming such a connection is mandatory in our case. Moreover, it is plausible that adenomyotic lesions may undergo carcinogenic changes within the myometrium, potentially progressing to a STUMP or even leiomyosarcoma, but this is a hypothesis requiring further molecular and histopathological validation.

In the case of our patient, even though the sequence of pathologies was unrolled in a short timeframe, we have no clear data to help us substantiate a common underpinning or link between them. However, we may speculate the presence of a disruptive background of the endometrial junctional zone, which may be only focal, leading to submucous myoma development and later adenomyoma occurrence.

## Data Availability

The original contributions presented in this study are included in the article. Further inquiries can be directed to the corresponding author(s).

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
