# Peer review of "STUMP Swiftly Followed by Large Adenomyoma in a Young Nulliparous Patient"

_diagnostics, 2025, doi:10.3390/diagnostics15233018_

Round 1
Reviewer 1 Report
Comments and Suggestions for Authors
I thank the authors for their work.
This case represents an exceptionally rare clinical presentation (STUMP (smooth muscle tumor of uncertain malignant potential) and large, heterogeneous, rapid-onset, and fast-growing adenomyosis), particularly given the patient’s young age, which posed a significant dilemma for all parties involved in the decision-making process. Performing a total abdominal hysterectomy with bilateral salpingo-oophorectomy (TAH-BSO) at such an age presented a considerable ethical and clinical challenge. The presence of a borderline ovarian tumor further compounded the complexity of management. It is recommended that Figure 1 be enhanced with annotated arrows to improve clarity and facilitate interpretation.
Author Response
Esteemed reviewer, thank you for the time and effort put into assessing our work. We are deeply grateful for you appreciation of our case, also from the point of view of the difficult decision-making process.
Please forgive me for being so bold as to say there are no clear annotations we can make in Figure 1, the stainigns are what they are, to the better understandig of pathologists, perhaps.
I will add below some refrences which all include similar STUMP immunohistochemistry and have no annotations on figures.
https://jmedicalcasereports.biomedcentral.com/articles/10.1186/1752-1947-5-214
https://link.springer.com/article/10.1007/s40944-022-00660-x
https://www.nature.com/articles/s41598-021-83711-1
Reviewer 2 Report
Comments and Suggestions for Authors
This Interesting Images, “STUMP swiftly followed by large adenomyoma in a young nulliparous patient,” presents an unusual clinical sequence of events: the occurrence of a smooth muscle tumor of uncertain malignant potential (STUMP) followed within nine months by a rapidly growing large adenomyoma in the same patient. The manuscript is clearly written, appropriately referenced, and clinically relevant. However, despite its novelty as a case description, the scientific depth, analytic rigor, and imaging documentation are insufficient for an international journal of this level without further refinement. Below are major and minor points to address before the manuscript can be considered suitable for publication.
- The authors claim this is the first reported case of sequential STUMP followed by a large adenomyoma. The authors should emphasize whether this represents a coincidental co-occurrence, a transformation continuum, or a shared pathogenic background (e.g., hormonal, molecular, or stromal remodeling mechanisms).
- The reported sequence (“STUMP removal → normal follow-up → adenomyoma after 9 months”) is intriguing but not fully substantiated. Key clinical intervals (exact time of imaging, follow-up schedule, hormonal therapy, or recurrence symptoms) are unclear.
Recommendation: Provide a chronological table summarizing interventions, imaging dates, and pathology results to clarify disease progression.
- Minor spelling and formatting issues: The article should definitely be checked!
“3 moths follow-up” → “3-month follow-up”
“rapidly evolving tumor size” → “rapidly enlarging uterine mass”
“half-yearly follow-ups” → “biannual follow-ups”
Author Response
Esteemed reviewer, thank you taking the time to read and evaluate our work, for your appreciation of our manuscript and for outlining our shortcomings. We have done our best to amemnd the article, you were quite right on every aspect and we are thankful for your suggestions. The changes inputed have been highlighted in blue.
Reviewer: The authors claim this is the first reported case of sequential STUMP followed by a large adenomyoma. The authors should emphasize whether this represents a coincidental co-occurrence, a transformation continuum, or a shared pathogenic background (e.g., hormonal, molecular, or stromal remodeling mechanisms).
Authors: Dear reviewer, we see your point and this is somehow part of our own reflections and even drive behing writing this article. However, in reality, it impossible to state this is a continuum, above the level of speculations. At the same time, it is odd to have such a sequence of patholologies occurring in a short period of time without some common ground that we cannot put our finger on.
We have added the following text at the end of the manuscript.
“In the case of our patient, even though the sequence of pathologies was unrolled in a short timeframe, we have no clear data to help us substantiate a common underpinning or link between them. However, we may speculate the presence of an endometrial junctional zone disruptive background, maybe only focal, leading to submucous myoma development and later adenomyoma occurrence.”
Reviewer: The reported sequence (“STUMP removal → normal follow-up → adenomyoma after 9 months”) is intriguing but not fully substantiated. Key clinical intervals (exact time of imaging, follow-up schedule, hormonal therapy, or recurrence symptoms) are unclear.
Recommendation: Provide a chronological table summarizing interventions, imaging dates, and pathology results to clarify disease progression.
Authors: Thank you for your suggestion, it is true, even with our description, it is still unclear how the events succeeded. We have been counseled by the editor that a table is not suitable for this type of paper and so we have provided Figure 2 as a timeframe of events. We hope this is both illustrative and acceptable from the journal layout point of view.
Reviewer: Minor spelling and formatting issues: The article should definitely be checked!
Authors: We have double checked the article for spelling errors and typos, as well as English language meaning, thank you.
Round 2
Reviewer 2 Report
Comments and Suggestions for Authors
Necessary arrangements have been made.